complexity

evolutionary game theory, cooperation, altruism, experiments

**Author for correspondence:**
C. Gracia-Lázaro
e-mail: cgracia@bifi.es

# Framing in multiple public goods games and donation to charities

F. Maciel Cardoso[1,3], S. Meloni[1,4], C. Gracia-Lázaro[1,2,3], A. Antonioni[5], J. A. Cuesta[1,3,5,6], A. Sánchez[1,3,5,6] and Y. Moreno[1,2,3,7]

[1]Institute for Biocomputation and Physics of Complex Systems, Universidad de Zaragoza, Zaragoza, Spain
[2]Department of Theoretical Physics, Faculty of Sciences, Universidad de Zaragoza, Zaragoza, Spain
[3]Unidad Mixta Interdisciplinar de Comportamiento y Complejidad Social (UMICCS), Spain
[4]IFISC, Institute for Cross-Disciplinary Physics and Complex Systems, Palma de Mallorca, Spain
[5]Interdisciplinary Group of Complex Systems (GISC), Department of Mathematics, Universidad Carlos III de Madrid, Spain, and [6]UC3M Santander Institute of Big Data (IBiDat), Universidad Carlos III de Madrid, Leganés, Spain
[7]ISI Foundation, Turin, Italy

FMC, 0000-0002-2789-9250; SM, 0000-0001-6202-3302; CG-L, 0000-0002-9769-8796; AA, 0000-0002-5788-3348; JAC, 0000-0001-9890-9367; AS, 0000-0003-1874-2881; YM, 0000-0002-0895-1893

The vast amount of research devoted to public goods games has shown that contributions may be dramatically affected by varying framing conditions. This is particularly relevant in the context of donations to charities and non-governmental organizations. Here, we design a multiple public goods experiment by introducing five types of funds, each differing in the fraction of the contribution that is donated to a charity. We found that people contribute more to public goods when the associated social donations are presented as indirect rather than as direct donations. At the same time, the fraction of the donations devoted to charity is not affected by the framing. We have also found that, on average, women contribute to public goods and donate to charity significantly more than men. These findings are of potential interest to the design of social investment tools, in particular for charities to ask for better institutional designs from policy makers.

## 1. Introduction

The number and economic relevance of charities and non-governmental organizations has rapidly grown in the last few decades. For instance, more than 1.5 million non-profits were registered with the US IRS in 2019, contributing around 5.5% to

the US GDP [1]; adjusted for inflation, total giving increased by 2.4% from 2018 to 2019. In 2018/2019, there were 166 592 voluntary organizations in the UK; 19.4 million people volunteered at least once a year during 2018/2019 through a group, club or organization. In June 2019, the voluntary sector employed 909 088 people, which constitutes almost 3% of the total UK workforce [2]. This growth has been fuelled by the subsidies of many governments around the world, either by transferring funds directly to organizations or through tax deduction policies to donors [3–10]. At the same time, more than 1 billion people give money to charities [11]. In view of this volume of activity, philanthropy and voluntary contributions to charities have aroused the interest of a growing number of researchers in the last decades [12,13], leading to theoretical models [14], qualitative research [15], and experimental studies on the economics of charity [16], fundraising events [17], different forms of fundraising [18], and the effect of status [19], lead donors [20], rebates [21], subsidies [22], and message framing [23] on charitable giving.

Secondly, when studying altruistic behaviour in humans, gender differences deserve special attention. Empirical evidence suggests that women give more to charities than men [24]. Socio-cultural and evolutionary theories predict sex-differentiated behaviour [25], although they often disagree on how men and women will behave in specific circumstances. Socio-cultural theory stresses the role of cultural stereotypes [26] whereas evolutionary theory explains sex behavioural differences as adaptations [27]. Particularly, both theories agree on the existence of behavioural differences with respect to cooperation or altruism. Many experiments have been conducted to assess these differences, and in general, women show higher levels of cooperation and altruism than men [28–31], although other studies show that gender does not affect these traits [32,33]. The current consensus is that the outcome depends on the context [25], especially on how it aligns with common gender stereotypes [26].

One of the most frequently used frameworks to experimentally address donations to charities is that of public good games (PGGs) [34]. The representation of donations to charities through a PGG is far from perfect, but approximate enough to have been considered often in the literature [35–37]. In this context, the research question we address in this paper focuses on the effects of framing on both contributions to PGGs and donations to charities (for a recent review on framing in PGGs, the reader is referred to [38]). In order to compare the effectiveness of different fundraising schemes, we have carried out an experiment involving contributing to multiple PGGs simultaneously. In this type of experiment, subjects can choose between two or more common pots to allocate their endowments, and the choices made by them are used to assess the effects of different framings [39–42] or the appearance of behavioural spillovers between games [43–45]. Our work resorts to this experimental set-up but deviates from their research questions, insofar as our purpose is to better understand the willingness to donate as a function of the stipulated donation as discussed below. Note that, in this work, the charity donations are done through public goods contributions, but not directly from the players' endowments. As explained below, albeit the altruism is an indirect measure, this set-up allows us to measure simultaneously cooperative behaviour and altruism, and the effect of framing in both behaviours.

In our case, we have compared two distinct methods for raising funds: direct versus indirect donations. To this end, we have devised a special-purpose PGG with two different treatments: a first set-up involving an explicit social fee, or tax (*Direct-Donation*, henceforth DD), and another one involving an implicit social fee (*Indirect-Donation*, henceforth ID). As we will see below, our set-up allowed us to simultaneously measure two variables: the contributions to public goods and the amounts donated to charity. Regarding those donations, the very existence of a direct self-benefit precludes measuring altruism, and, therefore, we have given the subjects the chance to contribute to several PGGs, which differed in the fraction of the benefit that goes to charity. Furthermore, the existence of funds with different social taxes enables us to study the pattern of contributions and their corresponding framing effect. In this regard, our work is not unrelated to the extensive literature on nudging in PGGs and related prosocial situations [46–52], and in particular in the context of donations [53,54].

Our experiment provides several relevant conclusions concerning how people respond to framings intending to increase contributions with a social impact channeled through charities. Players play two phases consecutively. In the *Keep in the Pocket* (KP) phase, they can decide their total contribution to the public goods, saving the rest of the round endowment. In the *Forced Contribution* (FC) phase, they have to distribute all their endowments into the different public goods (see *Experimental design* for details). Therefore, our experimental design allows us to extract information about willingness to donate— elicited in the KP and measured as the fraction of money allocated to public goods, which we refer to as contribution—and also about the donations themselves, measured in the FC phase as the fraction of money that actually goes to charity, which we call donation. As we discuss below, our results show that the ID treatment gives rise to higher contributions to public goods than DD. On the other hand, the fraction of the investments donated to charity, i.e. the donation, is not affected by the framing. As a

consequence, the combination of these two effects leads to a higher amount donated in the ID treatment (same fraction of a larger contribution). An additional relevant result touches upon gender influence, as we have found that women contribute to public goods and donate to charity more than men. All these findings may have implications of interest for the design of socially responsible investing strategies.

# 2. Results

## 2.1. Experimental design

Figure 1 shows a schematic of the experimental set-up. Experiments were conducted on groups of 10 participants. Each group played an iterated PGG with five funds, which differed in the fraction of profit donated to charity (0%, 5%, 10%, 15%, 20%, respectively). In a standard PGG, participants contribute to a common pot, and the total of the pot is multiplied by the so-called multiplication factor, being subsequently equally distributed among all participants irrespective of their contribution. In every round, subjects were given 100 experimental currency units (hereafter, ECU) which they could distribute among the five funds at will. Donations enter the experiment by having some money taken from the subjects' earnings and sending it to a charity (specifically, Médecins Sans Frontières/ Doctors Without Borders, see below).

In the above framework, our goal is to compare two natural approaches to implement donations in a PGG scenario: donations coming from taxes on the contributions or coming from decreases in the profitability. To study the effects among these two framings, we split participants into two treatments. In the *direct-donation* (DD) treatment, once the contribution of the round was made, a fraction to be donated to charity was removed from each fund and the remaining amount was multiplied by 1.5 and equally distributed among all participants. The fraction destined to the charity was 0% (no donation whatsoever), 5%, 10%, 15% or 20%, according to the chosen fund, whereas the multiplication factor was the same for all the funds. Conversely, in the *indirect-donation* (ID) treatment, subjects were informed that the experimenters (the 'bank') would make the donation, which was implemented by a decrease in the multiplication factor and hence of the money received by the subject in return for his investment. In order to ensure that each fund would yield exactly the same payoff in any of the two set-ups, the multiplication factors used in the ID treatment were 1.5, 1.425, 1.35, 1.275, 1.2, in one-to-one correspondence to the donation rates of the DD treatment. The mathematical equivalence in terms of payoffs and donations between treatments DD and ID is shown in the section Methods.

Moreover, to elicit how much money people actually donate and how this affects the choice of fund or funds, every group played two phases in a row: one in which subjects had to contribute all 100 ECUs (*Forced-Contribution*, henceforth FC), and a second one in which they were allowed to keep as much of those 100 ECUs as they wished and contribute the rest (*Keep-in-the-Pocket*, henceforth KP). Each one of these two phases consisted of 20 rounds, and subjects played these two phases consecutively. The number of rounds was unknown by the participants—who were only informed that the experiment would last for an indefinite number of rounds.

Although all participants played both FC and KP, the order of both phases was not the same for all the groups: half of the groups played first the FC phase (first forced contribution order, henceforth FFC) and the rest of the groups played first the KP (first keep in the pocket order, henceforth FKP). Note that the order may play a role in framing: FFC participants are only concerned initially with their fund options, whereas FKP ones first have to decide between saving or contributing since the very beginning, ending up later with the distribution decision only. Table 1 reports the number of subjects for each treatment and first played phase combination, for a total of 120 involved participants.

Accumulated payoffs could not be reinvested: the maximum amount that subjects could contribute every round was the 100 ECUs that they received afresh at the start of the round. In every round, the information about the characteristics of the fund was of course available to them: subjects were shown the fraction destined to charity by each fund (DD treatment) or the corresponding multiplying factors (ID treatment). Before the experiment, the researchers verbally informed the subjects about the destination of the charity donations: Médecins Sans Frontières (Doctors Without Borders). At the end of each round, subjects were informed about the total amount contributed to each fund among all the participants of their group, but not about each individual contribution. At the end of the experiment, each player received the payoffs accumulated along all the rounds played converted to euros, plus a fixed show-up fee. See the Methods section for further details and §2 of electronic supplementary material for the full instructions used in the experiment and screenshots of the interface.

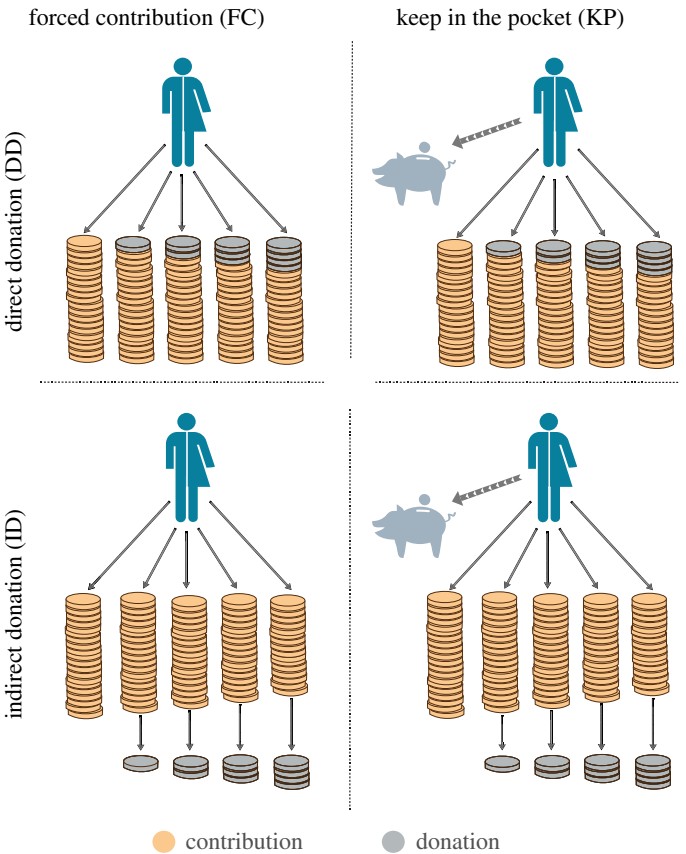

**Figure 1.** Experimental set-up. Participants played a PGG adaptation wherein they could contribute to five different funds. In the *Direct Donation* (DD) treatment, all the funds had the same profitability; each fund involved a different charitable donation rate to be deducted from the contributions (player pays). In the *Indirect Donation* (ID) set-up, funds involved different profitabilities according to the donation rate; the donation was made by the experimenters, which was implemented by a decrease in profitability (bank pays). Funds are designed such that associated benefits and donations are the same in both treatments, and participants were randomly assigned to one of them. In each treatment, participants played two consecutive phases: in *Forced Contribution* (FC), participants were required to contribute all their endowment to the available funds; in *Keep in the Pocket* (KP), participants chose how much to contribute to the funds, keeping the remaining for them. Accordingly, there were two cohorts: half of the participants played first the FC (FFC order), while the other half played first the KP (FKP).

**Table 1.** The number of participants in each cohort. Each participant was designated to one of two treatments: *Indirect Donation* (ID) or *Direct Donation* (DD). Furthermore, all participants played two phases, namely, *Forced Contribution* (FC) and *Keep in the Pocket* (KP).

|  | indirect donation | direct donation |
| --- | --- | --- |
| first forced contribution | 30 | 30 |
| first keep in the pocket | 30 | 30 |

## 2.2. Data analysis

We measure the effectiveness of the two different fundraising methods, DD and ID, through the differences in contributions and donations. Contributions to public goods are measured in the KP phase as the fraction of the 100 ECUs that a player contributes in all five funds, while donations can be measured in FC as the fraction of the 100 ECUs that goes to charity. Just for clarity, we remind the reader that in FC the entire endowment must go to funds and the only choice subjects can make is how it is distributed among the different funds (and their corresponding donation rates).

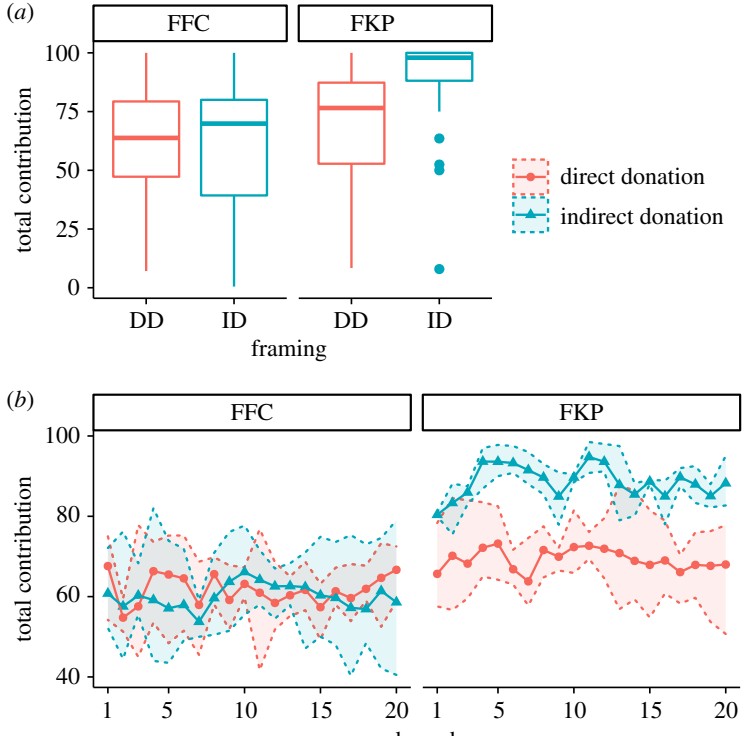

**Figure 2.** Contributions to public goods are sensitive to framing only in the FKP order. (*a*) Boxplot of the average total contribution to PGGs by subject (i.e. average of the individual contributions accumulated over all the rounds). Each colour corresponds to a treatment: one involving an explicit social fee (*Direct Donation*, DD), and the other one involving an implicit social fee (*Indirect Donation*, ID). In the left panel (FFC), individuals played first the FC phase in which they had to allocate all their endowments into the different public goods and, subsequently, the KP phase. In the right panel, subjects (FKP) played the KP phase first, in which they chose how much to contribute to the funds, saving the remaining. In both panels, contributions were measured in the KP phase. The lower and upper hinges correspond to the first and third quartiles. The upper (resp. lower) whisker extends from the hinge to the largest (resp. smallest) value no further than $1.5 \times$ IQR (interquartile range) from the hinge. (*b*) Groups' average contribution to PGGs at each round for each order (FFC and FKP). The shaded area corresponds to 0.95 bootstrapped confidence interval.

### 2.2.1. Contributions

The results of the contribution, i.e. the fraction of money invested in the public goods when subjects could keep all or part of it in their pockets, are shown in figure 2. Boxplots of panel A show the average total contribution by subject averaged over the 20 rounds of the KP phase, while panel B displays the evolution over time of the averaged group contribution. As shown in both panels, the FKP order exhibits an influence of framing on contributions that is not present in the FFC. To evaluate the significance of this dependence, we have implemented a random-effects model [55,56]. The details of this analysis are described in section Methods below. The results of the random-effects model are displayed in table 2. The analysis shows that there is a framing effect on contributions, as participants from different treatments do not contribute the same amount. Participants in the ID treatment contribute significantly more, although this effect is only observed when they begin the experiment playing KP. Furthermore, it turns out that women contribute significantly more than men. Therefore, our results show that participants tend to contribute more when the donation is done by an external agent instead of directly extracting the amount out of their earnings. Nevertheless, this effect is only observed when they have not previously participated in a phase of forced contributions.

### 2.2.2. Donations

Regarding donations to charity, although present in both FC and KP phases, measuring them in the FC phase allows removing the effect of the contributions to the public goods. The results of the contributions to charity in FC are shown in figure 3. Boxplots of panel A display the average total donation by subject averaged over the 20 rounds of each phase and panel B shows the averaged group contribution as a

**Table 2.** Random-effects regression results (Wallace and Hussain estimator) with cluster robust standard errors at the individual level for the contributions to public goods. Column (1) refers to the model for subjects' contributions being DD the reference (equation (M1)). Column (2) refers to the model (1) after adding the FKP term to take into account the order plus an additional term for the interaction between the order and the treatment, being DD × FFC the reference (equation (M2)). Column (3) refers to the model (2) after adding a $W$ term for the gender, being the reference a male subject playing DD × FFC (equation (M3)). See section Methods for further details.

| | dependent variable: | | |
| | contribution | | |
| | (1) | (2) | (3) |
|---|---|---|---|
| indirect donation | 8.886* | −1.655 | −1.657 |
| | (5.051) | (7.369) | (7.133) |
| FKP | | 7.636 | 8.451 |
| | | (6.632) | (6.512) |
| woman | | | 12.186** |
| | | | (5.091) |
| indirect donation × FKP | | 21.089** | 21.088** |
| | | (9.332) | (9.078) |
| constant | 65.635*** | 61.817*** | 54.100*** |
| | (3.357) | (5.051) | (6.510) |
| observations | 2363 | 2363 | 2363 |
| $R^2$ | 0.017 | 0.115 | 0.146 |
| adjusted $R^2$ | 0.017 | 0.113 | 0.145 |
| $F$ statistic | 41.235*** | 305.333*** | 403.005*** |

Note: $^*p < 0.1$; $^{**}p < 0.05$; $^{***}p < 0.01$

function of the round number. Here, the donation is measured as the fraction of the 100 ECUs that goes to charity. Both panels suggest that there is no difference in donations between FC and KP phases regardless of the order they were played (either FFC or FKP).

To confirm the lack of framing effect in donations to charity, we have resorted to a random-effects model as in the case of contributions to public goods. The details of the analysis are described in the section Methods, and the results are displayed in table 3. The analysis confirms that, regarding donations to charity, there is neither difference between treatments nor order effects. Nonetheless, we still observe a robust finding, namely that women donate significantly more (i.e. contribute to funds with higher donation rate) than men.

Summarizing contributions and donations results, we have shown that the ID treatment gives rise to higher contributions to public goods than DD. As it turns out that the fraction of those investments donated to charity is not affected by the framing, the total amount donated is also higher in the ID treatment. Let us now turn to a more detailed analysis of how subjects distributed their donations among the different funds.

## 2.2.3. Distribution of contributions

The experiment was designed with five different funds with different social taxes to allow us to study the pattern of contributions and the effects of framing on it. In addition, in the case of a framing effect, being

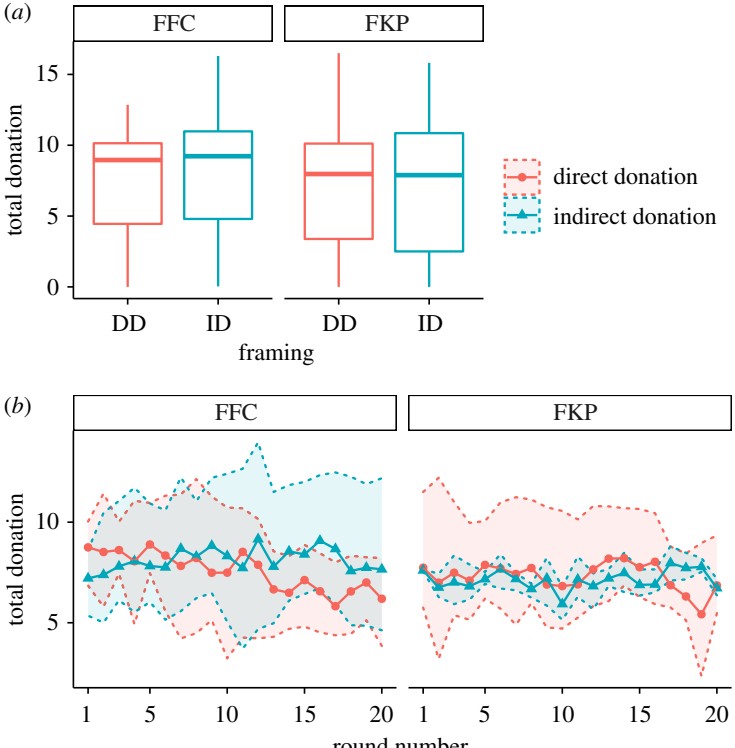

**Figure 3.** Total donations to charity in the FC phase. (*a*) Boxplot of the average total donation by subject (i.e. average of the individual donations accumulated over all the rounds). The lower and upper hinges correspond to the first and third quartiles. The upper (resp. lower) whisker extends from the hinge to the largest (resp. smallest) value no further than $1.5 \times$ IQR from the hinge. (*b*) Group averages at each round. The shaded area corresponds to 0.95 bootstrapped confidence interval.

able to extract a pattern in the contributions can help us to investigate the possible drivers behind the differences between the two framings.

In order to study the distribution of contributions in the five different funds, we have carried out regression analyses of the contributions in both FC and KP phases. The details of these analyses are described in the Methods section. Note that a different regression has been performed for each fund in a given phase.

Table 4 shows the results for the FC phase, wherein participants could only decide how to distribute their contributions. As shown, participants from different treatments are not more likely to contribute to one fund over another. The only significant effect is observed for gender, women being more likely to contribute a larger share to funds with a positive social tax while contributing less to the fund with no social tax. Thus, they end up donating to charity more than men.

Table 5 displays the result of the same regressions for the KP phase, wherein participants can decide the amount they contribute to public goods. Clearly, women also donate to charity more than men in this phase, as they are more likely to choose funds with a higher social tax. There is a higher contribution associated with participants playing the ID treatment in the FFC, nonetheless, this was expected as participants contributed more in total as shown in previous sections.

Therefore, as can be seen, differences in individual funds' contributions are a consequence of our previous findings, namely, being a woman or playing the ID treatment in order FFC, which is associated with higher contributions. We do not observe any additional effects, indicating that the donation rate itself, be it direct or indirect, was not an important factor for participants' choices.

## 3. Discussion

Let us now discuss the observed differences in contributions between the two frames, as well as the observed gender differences, for which we have explored two possible explanations. On the one hand, we considered the possibility of conditional contributions, i.e. of subjects reacting to the total amount

**Table 3.** Random-effects regression with cluster robust standard errors at the individual level for the donations to charity. Column (4) refers to the model for subjects' donations with DD as the reference (equation (M4)). In (5), two terms have been added to (4): the FKP term taking into account the order, plus an additional term for the interaction between the order and the treatment, being DD × FFC the reference (equation (M5)). Column (6) refers to the model (5) plus a W term for the gender, being a male subject playing DD × FFC the reference (equation (M6)). See section Methods for further details.

| | dependent variable: | | |
| | total donation | | |
| | (4) | (5) | (6) |
| --- | --- | --- | --- |
| indirect donation | 0.198 | 0.548 | 0.549 |
| | (0.803) | (1.002) | (0.963) |
| FKP | | −0.273 | −0.010 |
| | | (1.129) | (1.013) |
| women | | | 3.943*** |
| | | | (0.772) |
| indirect donation × FKP | | −0.700 | −0.701 |
| | | (1.601) | (1.438) |
| constant | 7.446*** | 7.583*** | 5.086*** |
| | (0.565) | (0.660) | (0.778) |
| observations | 2347 | 2347 | 2347 |
| $R^2$ | 0.0004 | 0.005 | 0.139 |
| adjusted $R^2$ | −0.0001 | 0.003 | 0.138 |
| F statistic | 0.546 | 10.542** | 379.479*** |

Note: *$p < 0.1$; **$p < 0.05$; ***$p < 0.01$.

contributed by the rest of their group, known to them as this information was provided after each round as indicated above. To check this hypothesis, we have performed regression analyses to evaluate possible differences between treatments with respect to the responses of subjects to the behaviour of the rest of the participants in their group, as well as differences in the conditional contribution between genders. The details of these analyses can be found in §5 of electronic supplementary material. Regression results indicate that participants do not condition their contribution to other participants' behaviour. There is neither evidence that men or women would react differently to this general trend, nor significant differences between different framings in this respect. In view of this, we have to exclude this possibility.

Another plausible explanation of the observed influence of framing on contributions relates to how information is presented to the subjects. In this regard, taxes are only shown to participants playing the DD treatment, the salient presence of taxes being the main difference between the two treatments. According to this explanation, participants in the ID treatments would react negatively to the tax while contributing. Conversely, subjects in the DD treatments are not affected by the reduction in the profitability to the same degree. Furthermore, the fact that the framing effect is observed only in the FKP order suggests that subjects that play first the FC phase are conditioned by this learning effect, being their contributions in the subsequent KP phase independent of the framing. We believe that this hypothesis is a probable cause for our results, but confirming it would require further research on how the saliency of the method of donation affects behaviour.

**Table 4.** Random effects regression with cluster robust standard errors at the individual level for the FC phase. Each column corresponds to a fund of a determined social tax, namely: 0, 5%, 10%, 15%, 20%, from left to right. The reference is a male subject playing DD × FFC. See section Methods for further details.

|  | dependent variable: $C_{itf}$ | | | | |
|---|---|---|---|---|---|
|  | 0 | 5% | 10% | 15% | 20% |
| ID | −3.774 | 3.521 | −3.229 | 0.047 | 3.439 |
|  | (7.048) | (3.343) | (2.418) | (2.339) | (3.052) |
| FKP | 2.813 | −1.925 | −3.435 | 1.632 | 0.910 |
|  | (7.122) | (1.928) | (2.438) | (3.641) | (3.585) |
| women | −25.784*** | 2.738 | 5.441*** | 5.199** | 12.411*** |
|  | (5.744) | (2.545) | (1.774) | (2.428) | (2.358) |
| ID × FKP | 1.132 | 1.236 | 4.611 | −3.437 | −3.529 |
|  | (10.292) | (4.389) | (3.175) | (4.400) | (4.677) |
| constant | 58.036*** | 11.673*** | 10.405*** | 10.228*** | 9.642*** |
|  | (6.121) | (2.154) | (2.584) | (2.020) | (2.541) |
| observations | 2347 | 2347 | 2347 | 2347 | 2347 |
| $R^2$ | 0.133 | 0.021 | 0.047 | 0.029 | 0.095 |
| adjusted $R^2$ | 0.131 | 0.020 | 0.045 | 0.027 | 0.093 |
| F statistic | 358.453*** | 51.261*** | 115.645*** | 70.157*** | 245.595*** |

Note: *$p < 0.1$; **$p < 0.05$; ***$p < 0.01$.

## 4. Conclusion

In this paper, we have used an experimental set-up based on a PGG modified to include a social responsibility factor, to show that framing will affect fund contributions depending on how the donation procedure is implemented. On the one hand, we have found that contributions are higher when the associated social donations are presented as indirect donations rather than as social taxes. On the other hand, the fraction of the contributions devoted to charity is not affected by the framing effect. This result is not unrelated to the work of Krieg & Samek [42], where they observe that a return of 20% of the contribution back to the donor increases significantly the contribution level, whereas recognition or sanctions have no effect. On the other hand, our conclusion is in line with that of Altmann et al. [54], who found that defaults may have both desired and undesired effects on the distribution of donations and also an influence on people's decisions, even if this influence might not be directly apparent in aggregate-level data (see also [53]). We have tentatively connected this effect to the explicit information about taxes, but further research would be needed to confirm that this is actually the cause. We have also found that, on average, women contribute to the public goods and donate to charity more than men, which is observed in some philanthropy contexts [57]. The implications of these findings are relevant for policy-makers in the design of socially responsible investing strategies and fair policies, e.g. when the government or a charity intends to promote socially responsible conducts, or compete successfully for the limited amount of funds available to the different charities. People are not only self-interested, nonetheless, but their likelihood of acting prosocially can also be influenced by the type of incentive and economic context [58]. In this regard, the results of Corazzini et al. [41] point to the relevance of avoiding miscoordination among donors by making particular options salient. We stress here that, when given several possibilities for donations, we have observed a more or less uniform distribution, and that might imply that neither of the possibilities receives funding enough for their

**Table 5.** Random effects regression with cluster robust standard errors at the individual level for the Keep in the Pocket phase. Each column corresponds to a fund of a determined social tax, namely: 0, 5%, 10%, 15%, 20%, from left to right.

| | dependent variable: $C_{itf}$ | | | | |
| --- | --- | --- | --- | --- | --- |
| | 0 | 5% | 10% | 15% | 20% |
| ID | −3.948 | 0.812 | −2.066 | −2.292 | 5.839* |
| | (5.711) | (1.646) | (1.776) | (1.882) | (3.079) |
| FKP | 0.017 | 2.295 | 0.857 | 0.846 | 4.434 |
| | (5.480) | (1.571) | (1.716) | (1.764) | (3.196) |
| women | −11.060** | 4.103** | 4.643*** | 5.156*** | 9.336*** |
| | (4.718) | (1.778) | (1.256) | (1.377) | (2.275) |
| ID × FKP | 11.976 | 5.853* | 4.935** | 5.068* | −6.757 |
| | (8.265) | (3.366) | (2.474) | (2.658) | (4.570) |
| constant | 31.870*** | 4.616*** | 5.227*** | 5.906*** | 6.479*** |
| | (5.665) | (1.414) | (1.350) | (1.639) | (2.050) |
| observations | 2363 | 2363 | 2363 | 2363 | 2363 |
| $R^2$ | 0.057 | 0.077 | 0.065 | 0.063 | 0.070 |
| adjusted $R^2$ | 0.056 | 0.076 | 0.063 | 0.061 | 0.069 |
| $F$ statistic | 143.432*** | 197.424*** | 162.838*** | 157.617*** | 178.595*** |

Note: $^*p < 0.1$; $^{**}p < 0.05$; $^{***}p < 0.01$.

needs. Our findings point then to the importance of restricting choices in case concentration of donations is required to help one or a few charities. In fact, this might also be relevant in a wider context, such as climate change mitigation, when the availability of many different actions for individuals trying to help mitigation might backfire because of similar miscoordination reasons. Finally, our results may be important for charities themselves, in so far as they suggest that different institutional frameworks may lead to receiving more money, allowing them to pick those that are most favourable to them or talk to policy-makers to promote the most suitable ones.

# 5. Methods

## 5.1. Summary of participant data

Participants were recruited from the volunteer pool of the Laboratory for Research in Experimental Economics (Lineex) of the University of Valencia, Spain. The volunteer pool is open to people of different ages, education level, and social status. The experiments were performed at Lineex lab, on February 18 and 1 March 2016. One hundred and twenty volunteers of different ages, education level, and social status were recruited for this experiment (48 men and 72 women; see demographic tables in electronic supplementary material for a more detailed description of the subjects and the groups). Participants were randomly distributed into 12 groups of 10 subjects regardless of their gender. They were seated in front of a computer, isolated from the other participants. Nobody knew the identity of the other members of his/her group. Participants' earnings ranged from 15.6 euros to 22.5 euros, with an average of 18.26 euros (including a show-up fee of 5 euros). All 120 participants were paid in cash at the end of each experiment. Payments to participants amounted to 2191 euros, while the total donation to charity was 88 euros. Each experiment lasted about 70 min.

## 5.2. Mathematical equivalence of treatments DD and ID

Let $z_i$ denote the total contribution to fund $i$ ($i = 1, 2, \ldots, 5$) among all the players in a given round. In the DD treatment, a fraction $x_i$ devoted to charity is taken from each fund and, subsequently, the remaining amount is multiplied by 1.5 and equally distributed among all participants. Thus, the amount of fund $i$ that will go to the NGO in that round will be $z_i x_i$, and every one of the 10 participants will receive a payoff $1.5 z_i (1 - x_i)/10$. As for the ID treatment, if $r_i$ and $x_i'$ are, respectively, the multiplication factor and the social fee of fund $i$, then the amount devoted to the NGO in that round will be $z_i x_i'$, and every one of the 10 participants will receive a payoff $z_i r_i /10$. Accordingly, for both treatments to involve the same payoffs and donations,

$$z(1 - x_i)1.5/10 = \frac{zr_i}{10}, \quad zx_i = zx_i',$$

for any total contribution $z$ to the fund. Taking $x_1 = 0\%$, $x_2 = 5\%$, $x_3 = 10\%$, $x_4 = 15\%$ and $x_5 = 20\%$, we get $r_1 = 1.5$, $r_2 = 1.425$, $r_3 = 1.35$, $r_4 = 1.275$, $r_5 = 1.2$ and $x_i' = x_i$ ($i = 1, \ldots, 5$).

## 5.3. Random-effects model

In this section, we describe the random-effects models [55,56] performed in this work. Regarding contribution to public goods, equation (M1) describes the model for subjects' contribution ($C_{it}$) at time $t$, given that the participant $i$ was playing the ID treatment ($ID_i$), taking contributions in DD as the reference.

$$C_{it} = \beta_0 + \beta_1 ID_i + u_{it}, \tag{M1}$$

where the error term $u_{it}$ is composed by an unobserved individual effect ($\alpha_i$), a time effect ($\lambda_t$), and an idiosyncratic error ($\epsilon_{it}$) which naturally is not correlated with the regressor:

$$u_{it} = \alpha_i + \lambda_t + \epsilon_{it}. \tag{M1.1}$$

The results of this model are shown in column (1) of table 2. Given that participants played in two different orders, we should add an order term to the model. Equation (M2) adds the $FKP_i$ term to indicate if participant $i$ started playing KP, as well as an interaction term between the order and treatment effect ($ID_i \times FKP_i$):

$$C_{it} = \beta_0 + \beta_1 ID_i + \beta_2 FKP_i + \beta_3 ID_i \times FKP_i + u_{it}. \tag{M2}$$

The results are shown in column (2) of table 2. Furthermore, as participants indicated their gender, we could use this information to analyse how it affects their contribution. The resulting model is given by equation (M3), wherein $W_i$ indicates if the participant was a woman, being men subjects the reference group. Given that this data come from a controlled randomized experiment, the independent variables do not correlate with the error term.

$$C_{it} = \beta_0 + \beta_1 ID_i + \beta_2 FKP_i + \beta_3 ID_i \times FKP_i + \beta_4 W_i + u_{it}. \tag{M3}$$

The results of this last model are shown in column (3) of table 2. The same analysis is performed for the donation to public goods in the FC phase considering the total donation ($D_{it}$). Accordingly, equations (M4), (M5) and (M6) are analogous versions of (M1), (M2) and (M3), respectively. The results are shown in table 3, where columns (4), (5) and (6) correspond, respectively, to the models described by equations (M4), (M5) and (M6).

$$D_{it} = \beta_0 + \beta_1 ID_i + u_{it} \tag{M4}$$

$$D_{it} = \beta_0 + \beta_1 ID_i + \beta_2 FKP_i + \beta_3 ID_i \times FKP_i + u_{it} \tag{M5}$$

and $$D_{it} = \beta_0 + \beta_1 ID_i + \beta_2 FKP_i + \beta_3 ID_i \times FKP_i + \beta_4 W_i + u_{it}. \tag{M6}$$

## 5.4. Regression analysis for the contributions' distribution

In this section, we describe the individual regressions performed to study whether the contributions to each fund differed between the different treatments. Equation (M7) describes the regression for fund $f$, where the dependent variable $C_{itf}$ corresponds to the amount contributed by subject $i$, at time $t$ to the fund $f$.

$$C_{itf} = \beta_0 + \beta_1 ID_i + \beta_2 FKP_i + \beta_3 ID_i \times FKP_i + \beta_4 W_i + u_{it} \tag{M7}$$

Results of the regressions for the FC (resp., KP) phases are shown in table 4 (resp., 5).

Ethics. All participants in the experiments reported in the main text signed an informed consent to participate. Besides, their anonymity was always preserved (in agreement with the Spanish Law for Personal Data Protection) by assigning them randomly a username that would identify them in the system. No association was ever made between their real names and the results. As is standard in socio-economic experiments, no ethical concerns are involved other than preserving the anonymity of participants. This procedure was checked and approved by the Viceprovost of Research of Universidad de Zaragoza and Aragon ethics research committee, license no. PI17/0196. The experiment was subsequently carried out in accordance with the approved guidelines.

Data accessibility. The datasets generated and analysed in the current study are available in the Open Science Framework repository, doi:10.17605/OSF.IO/UPN3Q [59].

Authors' contributions. S.M., C.G.-L., A.A., J.C., A.S. and Y.M. designed the research and the experimental framework; S.M., C.G.-L. and A.A. implemented the experiment; F.C., C.G.-L. and J.C. prepared the initial version of the manuscript; all authors analysed the data, performed the statistical analysis, revised and approved the final version of the manuscript.

Competing interests. We declare we have no competing interests.

Funding. This work was partially supported by MINECO (Spain) and FEDER funds through grant no. FIS2017-87519-P (Y.M.) and FJCI-2016-28276 (A.A.); by Ministerio de Ciencia, Innovación y Universidades/FEDER (Spain/UE) through grant no. PGC2018-098186-B-I00 (BASIC) (A.S. and J.C.); by Comunidad de Madrid under grant no. PRACTICO-CM and by Comunidad de Madrid/Universidad Carlos III de Madrid under grant no. CAVTIONS-CM-UC3M (A.S.); by Comunidad de Aragón (Spain) through grant no. E36-20R to FENOL (C.G.L. and Y.M.); by the EU through FET-Proactive Project MULTIPLEX (contract no. 317532, Y.M.) and FET-Proactive Project DOLFINS (contract no. 640772, C.G.L., Y.M. and A.S.), and by the Spanish State Research Agency and FEDER funds, through the María de Maeztu Program for Units of Excellence in R&D (MDM-2017-0711, S.M.) and under the PACSS grant no. (RTI2018-093732-B-C22, S.M.).

Acknowledgements. We thank Antonio Cabrales and Pablo Brañas-Garza for pointing out to us several important references related to this research.

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
