## [Peer Review File · Royal Society Open Science]

Review History

RSOS-202117.R0 (Original submission)

Review form: Reviewer 1

Is the manuscript scientifically sound in its present form?

Yes

Are the interpretations and conclusions justified by the results?

Yes

Is the language acceptable?

Yes

Do you have any ethical concerns with this paper?

No

Have you any concerns about statistical analyses in this paper?

No

Recommendation?

Accept with minor revision (please list in comments)

Comments to the Author(s)

My overall opinion of the article is favourable. The research question is clear, the experiments are well designed, I see no problem in the statistics, and the results and conclusions are properly linked to the obtained results. The data are public, and the methods are well explained. I think it is a work that deserves to be published. Said that, during its reading, I did find some elements that I think the authors could clarify or improve:

The first time I read the article, the introduction was confusing describing processes and results that are not detailed until much later in the article. I understand that you have to read the whole article to understand the details. Still, I think it is possible to add some clarifying sentences during the text itself beyond referring the reader to later information. I mean, to give an example (not the only one), when explaining the Keep in the Pocket phase, I think it would be useful to briefly describe the intuition of the process in the experiment; otherwise it little tricky to follow. One aspect that I think would also improve the understanding and limitations of the article is the issue of the representation of donations to charities through a PGG. As the authors themselves mention, this representation is by no means direct. I think it is useful to include at least one or two paragraphs explaining the selection and limitations of this representation beyond referring the reader to previous works that have used this framework. I believe that this choice has clear implications for the boundaries of the conclusions and should be mentioned in the paper.

The last aspect that has not been clear to me is whether the participants knew the specific charity to which they were going to donate. The text mentions that it was to Doctors Without Borders. My question is whether this element can be part of the game framing itself and with influence the results and whether it would not have been better to keep it anonymous (in other words, if the charity had been Alcoholics Anonymous, would the results have changed?) If this information was not given to the participants, my question is meaningless, but I would make it more straightforward in the text.

Otherwise, I find the work interesting and valuable.

Review form: Reviewer 2

Is the manuscript scientifically sound in its present form?

Yes

Are the interpretations and conclusions justified by the results?

Yes

Is the language acceptable?

Yes

Do you have any ethical concerns with this paper?

No

Have you any concerns about statistical analyses in this paper?

No

Recommendation?

Accept with minor revision (please list in comments)

Comments to the Author(s)

For varying framing conditions, here the authors study human behaviour of contributions and donations via a multiple public goods experiment by introducing five types of funds. They show that people tend to contribute more to public goods when the associated social donations are presented as indirect instead of direct donations. The effect of framing conditions as well as gender difference on donations are also explored. By combining experimental data (which is precious during the pandemic) and quantitative analysis, the results are convincing to me. The paper is well written and the experimental protocol is elucidated clearly. I only have some minor comments:

In Fig. 1, the fraction of contribution and donation in the cartoon is not proportional to the five funds designed, and a quantitative map between the cartoon and the number could be considered. Besides, at the moment, it is hard to distinguish DD and ID solely by the cartoon.

I am not sure whether participants are told in advance the number of rounds of the experiment, and this could be mentioned in the experimental design for either case.

In Sec. 2.2.1 and Fig. 2, does the total contribution refer to the fraction or the absolute number of contributions?

That is very interesting to show gender difference regarding the experiment. It is indeed one of the main contributions of the work. However, the main results especially the associated figures are shown in the SI, could authors consider to move that in the main text? This could favour readers who pay much time in the main content.

For the opening sentence of the Introduction, is it possible to acquire fresh data (2019 or 2018?) regarding the fraction of GDP for the number listed?

Decision letter (RSOS-202117.R0)

Dear Dr Gracia-Lázaro

On behalf of the Editors, we are pleased to inform you that your Manuscript RSOS-202117 "Framing in multiple public goods games and donation to charities" has been accepted for publication in Royal Society Open Science subject to minor revision in accordance with the referees' reports. Please find the referees' comments along with any feedback from the Editors below my signature.

Please submit your revised manuscript and required files (see below) no later than 7 days from today's (ie 04-Jan-2021) date. Note: the ScholarOne system will 'lock' if submission of the revision is attempted 7 or more days after the deadline. If you do not think you will be able to meet this deadline please contact the editorial office immediately.

Please note article processing charges apply to papers accepted for publication in Royal Society Open Science (<https://royalsocietypublishing.org/rsos/charges>). Charges will also apply to

papers transferred to the journal from other Royal Society Publishing journals, as well as papers submitted as part of our collaboration with the Royal Society of Chemistry (<https://royalsocietypublishing.org/rsos/chemistry>). Fee waivers are available but must be requested when you submit your revision (<https://royalsocietypublishing.org/rsos/waivers>).

on behalf of Professor Matjaz Perc (Associate Editor) and Pietro Cicuta (Subject Editor)
openscience@royalsociety.org

Reviewer comments to Author:
Reviewer: 1

Comments to the Author(s)

My overall opinion of the article is favourable. The research question is clear, the experiments are well designed, I see no problem in the statistics, and the results and conclusions are properly linked to the obtained results. The data are public, and the methods are well explained. I think it is a work that deserves to be published. Said that, during its reading, I did find some elements that I think the authors could clarify or improve:

The first time I read the article, the introduction was confusing describing processes and results that are not detailed until much later in the article. I understand that you have to read the whole article to understand the details. Still, I think it is possible to add some clarifying sentences during the text itself beyond referring the reader to later information. I mean, to give an example (not the only one), when explaining the Keep in the Pocket phase, I think it would be useful to briefly describe the intuition of the process in the experiment; otherwise it little tricky to follow. One aspect that I think would also improve the understanding and limitations of the article is the issue of the representation of donations to charities through a PGG. As the authors themselves mention, this representation is by no means direct. I think it is useful to include at least one or two paragraphs explaining the selection and limitations of this representation beyond referring the reader to previous works that have used this framework. I believe that this choice has clear implications for the boundaries of the conclusions and should be mentioned in the paper. The last aspect that has not been clear to me is whether the participants knew the specific charity to which they were going to donate. The text mentions that it was to Doctors Without Borders. My question is whether this element can be part of the game framing itself and with influence the results and whether it would not have been better to keep it anonymous (in other words, if the charity had been Alcoholics Anonymous, would the results have changed?) If this information was not given to the participants, my question is meaningless, but I would make it more straightforward in the text.

Otherwise, I find the work interesting and valuable.

Reviewer: 2

Comments to the Author(s)

For varying framing conditions, here the authors study human behaviour of contributions and donations via a multiple public goods experiment by introducing five types of funds. They show that people tend to contribute more to public goods when the associated social donations are

presented as indirect instead of direct donations. The effect of framing conditions as well as the gender difference on donations are also explored. By combining experimental data (which is precious during the pandemic) and quantitative analysis, the results are convincing to me. The paper is well written and the experimental protocol is elucidated clearly. I only have some minor comments:

In Fig. 1, the fraction of contribution and donation in the cartoon is not proportional to the five funds designed, and a quantitative map between the cartoon and the number could be considered. Besides, at the moment, it is hard to distinguish DD and ID solely by the cartoon.

I am not sure whether participants are told in advance the number of rounds of the experiment, and this could be mentioned in the experimental design for either case.

In Sec. 2.2.1 and Fig. 2, does the total contribution refer to the fraction or the absolute number of contributions?

That is very interesting to show gender difference regarding the experiment. It is indeed one of the main contributions of the work. However, the main results especially the associated figures are shown in the SI, could authors consider to move that in the main text? This could favour readers who pay much time in the main content.

For the opening sentence of the Introduction, is it possible to acquire fresh data (2019 or 2018?) regarding the fraction of GDP for the number listed?

===PREPARING YOUR MANUSCRIPT===

Your revised paper should include the changes requested by the referees and Editors of your manuscript. You should provide two versions of this manuscript and both versions must be provided in an editable format:
 one version identifying all the changes that have been made (for instance, in coloured highlight, in bold text, or tracked changes);
 a 'clean' version of the new manuscript that incorporates the changes made, but does not highlight them. This version will be used for typesetting.

If you have been asked to revise the written English in your submission as a condition of publication, you must do so, and you are expected to provide evidence that you have received language editing support. The journal would prefer that you use a professional language editing service and provide a certificate of editing, but a signed letter from a colleague who is a native speaker of English is acceptable. Note the journal has arranged a number of discounts for authors

using professional language editing services
(<https://royalsociety.org/journals/authors/benefits/language-editing/>).

===PREPARING YOUR REVISION IN SCHOLARONE===

-- If you have uploaded ESM files, please ensure you follow the guidance at <https://royalsociety.org/journals/authors/author-guidelines/#supplementary-material> to include a suitable title and informative caption. An example of appropriate titling and captioning may be found at https://figshare.com/articles/Table_S2_from_Is_there_a_trade-

off_between_peak_performance_and_performance_breadth_across_temperatures_for_aerobic_sc
ope_in_teleost_fishes_/3843624.

Author's Response to Decision Letter for (RSOS-202117.R0)

See Appendix A.

RSOS-202117.R1 (Revision)

Review form: Reviewer 1

Is the manuscript scientifically sound in its present form?

Yes

Are the interpretations and conclusions justified by the results?

Yes

Is the language acceptable?

Yes

Do you have any ethical concerns with this paper?

No

Have you any concerns about statistical analyses in this paper?

No

Recommendation?

Accept as is

Comments to the Author(s)

I think that the authors have improved the paper and in my opinion it can be published in the present form

Review form: Reviewer 2

Is the manuscript scientifically sound in its present form?

Yes

Are the interpretations and conclusions justified by the results?

Yes

Is the language acceptable?

Yes

Do you have any ethical concerns with this paper?

No

Have you any concerns about statistical analyses in this paper?

No

Recommendation?

Accept as is

Comments to the Author(s)

I have no further comments.

Decision letter (RSOS-202117.R1)

Dear Dr Gracia-Lázaro,

It is a pleasure to accept your manuscript entitled "Framing in multiple public goods games and donation to charities" in its current form for publication in Royal Society Open Science. The comments of the reviewer(s) who reviewed your manuscript are included at the foot of this letter.

You can expect to receive a proof of your article in the near future. Please contact the editorial office (openscience@royalsociety.org) and the production office (openscience_proofs@royalsociety.org) to let us know if you are likely to be away from e-mail contact – if you are going to be away, please nominate a co-author (if available) to manage the proofing process, and ensure they are copied into your email to the journal.

on behalf of Dr Feng Fu (Associate Editor) and Pietro Cicuta (Subject Editor)
openscience@royalsociety.org

Reviewer comments to Author:

Reviewer: 1

Comments to the Author(s)

I think that the authors have improved the paper and in my opinion it can be published in the present form

Reviewer: 2

Comments to the Author(s)

I have no further comments.

Appendix A

Reviewer: 1

Comments to the Author(s)

My overall opinion of the article is favourable. The research question is clear, the experiments are well designed, I see no problem in the statistics, and the results and conclusions are properly linked to the obtained results. The data are public, and the methods are well explained. I think it is a work that deserves to be published.

Reply: We thank the Reviewer for his/her valuable and positive report, which has contributed to improving our manuscript. Please, find below our responses to the points raised.

Said that, during its reading, I did find some elements that I think the authors could clarify or improve:

The first time I read the article, the introduction was confusing describing processes and results that are not detailed until much later in the article. I understand that you have to read the whole article to understand the details. Still, I think it is possible to add some clarifying sentences during the text itself beyond referring the reader to later information. I mean, to give an example (not the only one), when explaining the Keep in the Pocket phase, I think it would be useful to briefly describe the intuition of the process in the experiment; otherwise it little tricky to follow.

Reply: We thank the reviewer for this comment. Following the Reviewer recommendation, in the revised version of the manuscript, we have briefly explained Keep in the Pocket and Forced contribution phases when they first appear. We have added the following sentence in the introduction:

“Players play two phases consecutively. In the Keep in the Pocket (KP) phase, they can decide their total contribution to the public goods, saving the rest of the round endowment. In the Forced Contribution (FC) phase, they have to distribute all their endowments into the different public goods (see Experimental design for details).”

One aspect that I think would also improve the understanding and limitations of the article is the issue of the representation of donations to charities through a PGG. As the authors themselves mention, this representation is by no means direct. I think it is useful to include at least one or two paragraphs explaining the selection and limitations of this representation beyond referring the reader to previous works that have used this framework. I believe that this choice has clear implications for the boundaries of the conclusions and should be mentioned in the paper.

Reply: We thank the reviewer for pointing this out. We have added the following text in the Introduction:

“Note that, in this work, the charity donations are done through public goods contributions, but not directly from the players' endowments. As explained below, albeit the altruism is an indirect measure, this setup allows us to measure simultaneously cooperative behavior and altruism, and the effect of framing in both behaviors.”

The last aspect that has not been clear to me is whether the participants knew the specific charity to which they were going to donate. The text mentions that it was to Doctors Without Borders. My question is whether this element can be part of the game framing itself and with influence the results and whether it would not have been better to keep it anonymous (in other words, if the charity had been Alcoholics Anonymous, would the results have changed?) If this information was not given to the participants, my question is meaningless, but I would make it more straightforward in the text.

Reply: We thank the Reviewer for this comment. We have specified in the revised version that participants were informed aloud before starting. As the Reviewer suggest, this information is part of the framing. We cannot infer the results if that information had been kept hidden or if the donations would have had another destination (even chosen by the participants). This question remains open, and we intend to investigate it in future work.

In the revised version, it reads as follows:

“Before the experiment, the researchers verbally informed the subjects about the destination of the charity donations: Médecins Sans Frontières (Doctors Without Borders).”

Otherwise, I find the work interesting and valuable.

Reply: Thank you very much for your positive and valuable comments.

Reviewer: 2

Comments to the Author(s)

For varying framing conditions, here the authors study human behaviour of contributions and donations via a multiple public goods experiment by introducing five types of funds. They show that people tend to contribute more to public goods when the associated social donations are presented as indirect instead of direct donations. The effect of framing conditions as well we gender difference on donations are also explored. By combining experimental data (which is precious during the pandemic) and quantitative analysis, the results are convincing to me. The paper is well written and the experimental protocol is elucidated clearly.

Reply: We thank the Reviewer for his/her valuable and positive report, which has contributed to improving our manuscript. Please, find below our responses to the points raised.

I only have some minor comments:

In Fig. 1, the fraction of contribution and donation in the cartoon is not proportional to the five funds designed, and a quantitative map between the cartoon and the number could be considered. Besides, at the moment, it is hard to distinguish DD and ID solely by the cartoon.

Reply: We thank the reviewer for this pointing this out. Following his/her suggestion, in Figure 1, we have made the contributions proportional to the actual ones corresponding to the five funds the five funds. Additionally, we have improved the text to define more accurately the experimental setup.

I am not sure whether participants are told in advance the number of rounds of the experiment, and this could be mentioned in the experimental design for either case.

Reply: We thank the reviewer for this suggestion. In the previous version of the manuscript, that information was given in Methods. For the sake of clarity, we have moved it to the Experimental design section. It reads as follows:

“Each one of these two phases consisted of 20 rounds, and subjects played these two phases consecutively. The number of rounds was unknown by the participants---who were only informed that the experiment would last for an indefinite number of rounds.”

In Sec. 2.2.1 and Fig. 2, does the total contribution refer to the fraction or the absolute number of contributions?

Reply: We thank the Reviewer for this observation. In Figures 2 and 3, Panel A refers to individual contributions accumulated over all the rounds; Panel B refers to group contributions (i.e., the summation over all the players in a game) per round. We have clarified it in the revised manuscript.

That is very interesting to show gender difference regarding the experiment. It is indeed one of the main contributions of the work. However, the main results especially the associated figures are shown in the SI, could authors consider to move that in the main text? This could favour readers who pay much time in the main content.

Reply: Thank you for this observation. We agree with the Reviewer that gender differences are one of the main results of this work. We have highlighted it in the text to emphasize this issue and attract the readers' attention. These results are shown through the random-effects regressions and the statistical analysis displayed in the tables.

For the opening sentence of the Introduction, is it possible to acquire fresh data (2019 or 2018?) regarding the fraction of GDP for the number listed?

Reply: We thank the reviewer for pointing this out. In the revised version, we have updated the data with the latest available (2020 reports).

Thank you very much for your positive and valuable comments.